# A Fairness-Aware Peer-to-Peer Decentralized Learning Framework with Heterogeneous Devices

**Zheyi Chen \*, Weixian Liao \***, **Pu Tian, Qianlong Wang** **and Wei Yu \***

Department of Computer & Information Sciences, Towson University, Towson, MD 21252, USA;
ptian1@students.towson.edu (P.T.); qwang@towson.edu (Q.W.)
\*   Correspondence: zchen12@students.towson.edu (Z.C.); wliao@towson.edu (W.L.); wyu@towson.edu (W.Y.)

**Abstract:** Distributed machine learning paradigms have benefited from the concurrent advancement of deep learning and the Internet of Things (IoT), among which federated learning is one of the most promising frameworks, where a central server collaborates with local learners to train a global model. The inherent heterogeneity of IoT devices, i.e., non-independent and identically distributed (non-i.i.d.) data, and the inconsistent communication network environment results in the bottleneck of a degraded learning performance and slow convergence. Moreover, most weight averaging-based model aggregation schemes raise learning fairness concerns. In this paper, we propose a peer-to-peer decentralized learning framework to tackle the above issues. Particularly, each local client iteratively finds a learning pair to exchange the local learning model. By doing this, multiple learning objectives are optimized to advocate for learning fairness while avoiding small-group domination. The proposed fairness-aware approach allows local clients to adaptively aggregate the received model based on the local learning performance. The experimental results demonstrate that the proposed approach is capable of significantly improving the efficacy of federated learning and outperforms the state-of-the-art schemes under real-world scenarios, including balanced-i.i.d., unbalanced-i.i.d., balanced-non.i.i.d., and unbalanced-non.i.i.d. environments.

**Keywords:** decentralized learning; learning fairness; heterogeneity

## 1. Introduction

With the advance and deployment of the Internet of Things (IoT), a variety of smart devices (smart phone, wearable device, wireless cameras, etc.) have significantly risen in the past decade, driving the evolution of smart-world IoT systems [1,2]. Smart devices with multiple sensors and computing components support various functions, including collecting raw data samples, processing computing tasks, transmitting data to cloud servers and other smart devices, among others.

Distributed machine learning has shown great potential to enhance the performance of smart-world systems. In particular, federated learning, one of the representative distributed machine learning frameworks, aims to train a global learning model with a central parameter server and a group of local smart devices. Compared to the centralized learning framework, federated learning utilizes the on-device data samples at the edge of a network [3]. It is composed of two steps iteratively: (i) *model aggregation*, in which the central parameter server collects the local training models for global aggregation, and (ii) *model update*, in which local learners, in each iteration, train the machine learning model on the local datasets [4]. Federated learning prevents private data leakage from a public data center.

Despite the benefits, there are a number of challenges for deploying federated learning frameworks under real-world scenarios. First, it demands a central parameter server to organize the local training process, the existence of which raises security risks. For example, the adversary could adopt a fine-tuning approach to craft some malicious updates released

from the central parameter server to compromise the entire local training group [5]. Moreover, federated learning frameworks using a central parameter server represent a single point of failure. The training process can be disrupted and/or terminated if the adversary compromises the central parameter server [6]. Furthermore, traditional federated learning frameworks overlook the learning fairness issue over real-world scenarios. Particularly, the federated learning model is trained with statistically heterogeneous data samples because local datasets are generally sampled from different data distributions, contributing to a non-i.i.d. learning scenario. The training process with a weight averaging-based aggregation scheme amplifies the negative impact of heterogeneous issues [7]. For example, some local nodes with disproportionately large datasets may dominate the training process, skewing the overall performance of trained models.

With the aforementioned challenges, in this paper, we first design a generic framework to explore the problem space of deploying decentralized machine learning in IoT systems. As a case study, we focus on one specific problem defined in the space and propose a peer-to-peer decentralized learning approach while considering learning fairness. We consider decentralized learning as a promising solution that could benefit the majority of local learners without compromising privacy. As shown in Figure 1, compared to the federated learning framework, the decentralized learning framework removes the central parameter server, in which all of the local learners are allowed to find a learning pair in a peer-to-peer manner. The proposed decentralized learning approach enables both model aggregation and model updates to occur locally. When it comes to the learning model, the proposed decentralized learning approach supports diverse learning preferences rather than one global learning model.

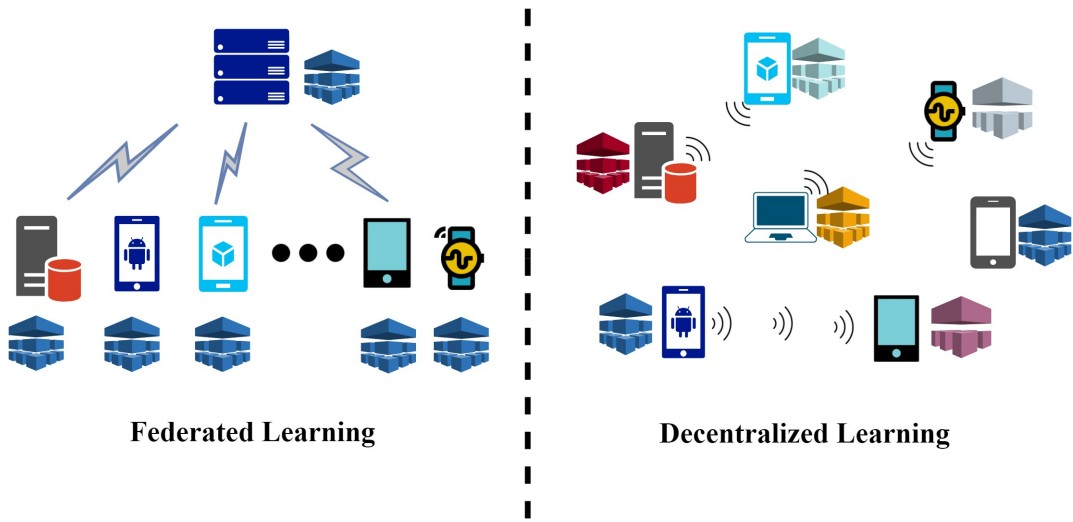

**Figure 1.** Distributed Learning vs. Decentralized Learning.

To tackle the learning fairness issue, we develop an adaptive aggregation scheme, which dynamically changes the weights of local updates based on the run-time learning performance. To achieve this goal, we propose a lightweight local performance metric into an aggregation scheme, where the proposed metric would adaptively update the weights of each learning model based on its local learning performance. Furthermore, the proposed scheme allows each local learner to train the learning model with its own preferences. To fulfill the various learning preferences of all of the clients, our approach achieves the overall optimization goal of the entire decentralized learning process. Last but not least, to validate the efficacy of our approach, extensive experiments are conducted over two public datasets.

The key contributions of this paper are summarized as follows:

- We design a generic framework with a three-dimensional problem space demonstrating the state-of-the-art research on deploying machine learning with IoT systems.

- We propose a fairness-aware adaptive aggregation scheme, which avoids small-group domination in the training process and further enhances the learning performance.
- Extensive experiments validate the efficacy of our approach over various scenarios.
- We discuss open issues in decentralized learning and outline potential research directions.

The remainder of this paper is organized as follows. We review related research in Section 2. In Section 3, we show the problem space of deploying distributed (decentralized) learning technologies in IoT. In Section 4, we present the problem formulation and the proposed fairness-aware approach. In Section 5, we show the performance comparison of our proposed approach with the other existing approaches. In Section 6, we discuss and outline several future research directions. Finally, we conclude the paper in Section 7.

## 2. Related Work

We now review the existing research efforts related to our study. While federated learning has received great attention recently, there are a number of open issues about its deployment in real-world scenarios, such as (i) data distribution (e.g., non-i.i.d., unbalanced data, and learning fairness issues); (ii) system security and reliability (e.g., security threats, communication efficiency, and others); and (ii)) learning structures (e.g., centralized, federated, decentralized). Next, we summarize the related works from these three aspects, respectively.

To address the learning fairness issues in a heterogeneous environment, Li et al. [7] proposed a federated learning framework with a fair resource allocation. A fairness metric ($q$-Fair coefficient) is parameterized by the variance of learning performance. Via the tuning $q$-Fair coefficient, the performance distribution of the proposed learning framework shows a balanced learning accuracy in the local training group. Ng et al. [8] proposed a multiplayer game under federated learning to study the action selection of federated learning participants in which various incentive mechanisms are involved. Lyu et al. [9] studied the collaborative fairness issue and proposed a reputation-based learning approach to release diverse models over the local training group. Moreover, Li et al. [10] investigated the fairness issues (i.e., performance distribution) and robustness (i.e., label poisoning, random updates, and model replacement) in the federated learning process. Likewise, Li et al. [10]) proposed a lightweight multitask learning approach to tackle the competing constraints of learning performance, fairness, and robustness in federated learning.

Some existing research works aim to solve the learning fairness problem of large-scale personalized federated learning [11,12]. For instance, Zhang et al. [12] proposed a learning approach, namely FedFomo, to achieve the model personalization. They designed a scheme to obtain an approximation of the optimal weight of a model combination for each local learner. Nonetheless, most of the aforementioned works have to deploy the proposed scheme with a central parameter server, which brings the extra computing cost and security concerns (e.g., single-point failure issue) to the learning process. Additionally, an unexpected backdoor would be spread to all of the local clients if the malicious updates can avoid the detection of a detection mechanism on the parameter server.

To consider the learning system security and reliability, Blum et al. [13] proposed a game-theoretic approach in federated learning to investigate the node behavior with various scenarios. The proposed learning scheme takes incentive-aware collaboration into account. Li et al. [14] introduced the blockchain technology into the federated learning process to address the potential security issues, namely the blockchain-based federated learning approach with committee consensus (BFLC). The proposed approach records the global models and the local model updates with the evaluation of the committee consensus mechanism. A trusted committee could be elected from three different methods, including random election, election by score, and multifactor optimization. Moreover, Chen et al. [4] proposed a zero-knowledge clustering-based scheme to mitigate adversarial attacks in federated learning environments. The proposed scheme is capable of not only alleviating the multiparty model poisoning attack during the training process but also

allowing the central parameters server to automatically modify the number of clusters in the detection stage.

Decentralized learning has been considered a promising paradigm to provide the flexibility over heterogeneous learning environments. Related to this direction, Kong et al. [15] investigated the performance gap between centralized learning and decentralized learning. Authors proposed a critical parameter, namely consensus distance, to guide the design of a decentralized learning process. It empirically shows that the performance gap between centralized learning and decentralized learning could be mitigated by controlling the consensus distance. Nonetheless, the effectiveness of the aggregation scheme remains an unsolved problem. Li et al. [16] proposed a transfer learning-based scheme to deal with the degraded learning performance and slow convergence. The mutual knowledge transfer algorithm was designed to improve the knowledge sharing over local training groups. In contrast, our work aims to find the fairness and diversity of local training groups. Likewise, Sun et al. [17] proposed the decentralized learning scheme with momentum to reduce the communication cost between local learners, in which the learning protocol allows clients to share local updates after multiple training iterations. Likewise, He et al. [18] proposed a group-knowledge transfer scheme to address the resource limitation issues on the edge service. To reduce the burden of the training process, the proposed scheme periodically transfers the knowledge of local groups to a large server-side learning model by adopting knowledge distillation.

## 3. A Distributed Learning Framework

We design a generic framework to explore the problem space and consider the state-of-the-art research efforts on deploying machine learning in IoT systems. Our proposed three-dimensional framework is shown in Figure 2, which considers the design space of machine learning in IoT systems from three perspectives (i.e., learning model architecture, resource distribution, and utility and requirements). Here, the $X$ dimension illustrates the machine learning architecture options in IoT systems (i.e., centralized, federated, and decentralized); the $Y$ dimension represents the resource distribution, which can be considered from a physical resource (e.g., homogeneous vs. heterogeneous devices) and data resource (e.g., i.i.d. vs. non-i.i.d. distributed data); and the $Z$ dimension shows utility and constraint requirements (e.g., model accuracy, efficiency, fairness, security, and privacy). Based on this framework, we can explore the research problems and map the existing research efforts into cubes in Figure 2, e.g., $< X$ (decentralized), $Y$ (heterogeneous, non-i.i.d.), $Z$ (security/reliability)$>$. In the following, we introduce the machine learning in IoT systems from the aspect of machine learning architectures, resource/data distribution, and security/reliability constraints.

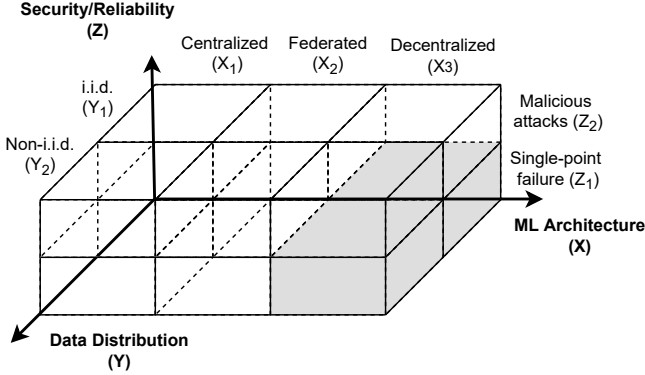

**Figure 2.** A Framework for Machine Learning in IoT.

### 3.1. Machine Learning Architectures in IoT

The conventional and straightforward strategy is to deploy machine learning in IoT systems via the centralized paradigm, where the system collects and processes data in one

central server. However, the centralized training might not work well in IoT systems as large amounts of data are normally generated by IoT devices deployed in different locations. Exchanging data between the IoT devices and a central server may pose significant communication overhead and further raises some data privacy and security implications. To address these issues, federated learning is proposed and shown in Figure 1. Here, each local data holder will not transmit the original data but compute the summarized information (e.g., gradients) first and then send the summarized one to the central server. After that, the central server will apply a selected aggregation function to consolidate the information from all of the local data holders so that the gradients of the learning model can be updated. As a further step, the decentralized learning fully eliminates the need for a center server to train the machine learning model. The federated and decentralized learning architectures will be better options for IoT systems because they are designed to train deep learning models with data stored in a decentralized or distributed manner.

### 3.2. Resource Distribution in IoT

#### 3.2.1. Non-i.i.d. Datasets

In IoT systems, the heterogeneity can pose challenges for machine learning [19]. Particularly, an IoT system has distributed clients and they usually collect non-i.i.d. datasets (amount, label, feature distribution, etc.). Due to the statistical differences, it is difficult to adopt an average aggregation mechanism to obtain a globally optimized model for all of the clients [20]. Zhao et al. [21] claimed that the performance impact of non-i.i.d. datasets could reach as high as 55 % and proposed a pre-shared dataset to control the divergence for federated learning. Ghosh et al. [22] proposed a cluster-based federated learning approach, in which the clients that hold data with similar distributions are grouped for aggregation. Likewise, Roy et al. [23] proposed a peer-to-peer distributed machine learning scheme, in which a node communicates with its peers and determines ones for aggregation based on the statistical distributions.

#### 3.2.2. Homogeneous vs. Heterogeneous Resources

The existing distributed learning systems (either federated or decentralized) generally focus on aggregating local models from distributed devices to generate and update a global model. Many of the existing works mostly assume homogeneous models where each party is homogeneous in terms of computing/communication resources, which may not be practical in IoT systems considering that the IoT devices might have uneven computing and data storage resources. Several research efforts tend to address this issue. For instance, Wu et al. [24] proposed a multiparty multiclass margin to calibrate heterogeneous local models in the system. Nonetheless, most of these works depend on a trusted centralized server for coordination and aggregation, which can possibly be subject to a single-point failure and cannot be used in a fully decentralized system. To the best of our knowledge, only a few efforts [25] attempted to develop the decentralized multiparty learning systems.

### 3.3. Security Constraints for Distributed Learning in IoT

For distributed learning architectures, a malicious user has the ability to inject poisoned data or compromised parameters to undermine the model [19,26]. In the data poisoning attack, the training datasets are manipulated, and the model is optimized with incorrect features [27,28]. For example, Jia et al. [29] demonstrated that the trained model for reading comprehension could be compromised with only a short sentence added to the sample. In distributed machine learning scenarios, the adversarial attacks could pose a serious risk due to the loose control of the participants [30]. Moreover, federated learning-based systems are prone to single-point failure [31]. In such a case, if the parameter server is down, the whole system will halt. To tackle such a problem, decentralized learning frameworks have been proposed [20,32]. Because the central server is not available and clients have no prior knowledge of other peers (i.e., distribution), the client itself has to guarantee its own performance during the aggregation process.

While the emerging decentralized learning solves the aforementioned issues in federated learning systems, it also raises other security concerns. Among them, Byzantine and Sybil attacks are two of the most common and critical attacks. Byzantine adversaries are the users who follow the system protocol and yet disseminate malicious information in the system, thereby misleading or even controlling the system performance. In the decentralized system, a Byzantine adversary may report bogus model information so that the normal model aggregation process in the system can be affected. Such a bogus model, being aggregated into the global or neighbor's model, leads the aggregated model to perform faulty results on some deliberately designed data samples. On the other hand, Sybil adversaries are the malicious users who create multiple identities to disrupt the system operations. In the decentralized learning system, a Sybil adversary may create multiple identities and submit a series of bogus information so that a certain Byzantine-tolerant aggregation method in a regular learning paradigm could be bypassed.

Decentralized machine learning frameworks can improve stability by reducing the dependency on the single node. In addition, a fully decentralized framework has to consider the heterogeneous problem (data distribution, unbalanced dataset, resource constraints, etc.). To ensure the performance, clients need to adopt an adaptive aggregation mechanism to establish the optimized model when receiving the information from its peers.

## 4. Fairness-Aware Decentralized Learning

We now introduce a case study that tackles one specific problem of decentralized learning in IoT systems, which belongs to subspace $(X_3, Y_1/Y_2, Z_1)$ highlighted in Figure 2. In the following, we first introduce the optimization problem in both the federated learning and decentralized learning. To solve the learning fairness issue, we then propose the adaptive aggregation scheme to improve the local learning performance over all of the local learning clients. For simplification, Table 1 lists the key notations in the paper.

**Table 1.** List of Key Notations.

| | |
|---|---|
| $\mathbb{W}^*$ | Group weight vector |
| $\mathbf{w}$ | Weight vector |
| $\mathbf{w}^*$ | Optimized parameters |
| $\mathbf{w}_i$ | Local learning parameters |
| $F(\cdot)$ | Loss function |
| $D$ | Local datasets |
| $\delta$ | Local update |
| $U$ | Update set |
| $t$ | Iteration index |

### 4.1. Problem Formulation

We denote the local learner set as $N = \{1, 2, \ldots, i, \ldots\}$ and denote the local datasets as $D_i$. Generally speaking, the central parameter server, in the federated learning framework, executes two major tasks: model updating and model aggregation. To generate a global learning model, federated learning adopts the weight averaging-based aggregation scheme [6,20]. The global optimization problem is formulated as:

$$\mathbf{w}_G^* \leftarrow arg\ min\ F(\mathbf{w}_G) = \sum_{i=1}^{|N|} P_i F_i(\mathbf{w}_G, D_i). \tag{1}$$

Here, denote $F(\cdot)$ as the loss function, $\mathbf{w}_G$ as the global learning model, and $P_i$ as the weight of local learner in the model aggregation. The federated learning framework provides a distributed way of exploiting the on-device data samples at the edge of network; however, it is challenging for distributed learning to be implemented in real-world scenarios. The central parameter server of federated learning causes the potential security risk, such as the single-point failure issue. The learning efficiency, in federated learning, would degrade

due to the diversity of local end-devices. To be specific, data transmissions (networking) and the efficiency of local model updates (with respect to computing) might bring the bottleneck of the global learning performance.

In decentralized learning, all local learners intend to find a learning pair to exchange its local model iteratively, known as peer-to-peer manner. Both model update and model aggregation are deployed on local learners. The objectives of decentralized learning differ from federated learning, aiming to train a number of local learning models with users' preferences. Mathematically, the optimization goal can formulated as:

$$\mathbb{W}^* \left\langle \mathbf{w}_1 \cdots \mathbf{w}_{|N|} \right\rangle \leftarrow arg\ min_{\mathbf{w}_i \in \mathbb{R}^d} \sum_{i=1}^{|N|} F_i(\mathbf{w}_i, D_i), \tag{2}$$

where $\mathbf{w}_i$ is the $i's$ local learning model. The nature of decentralized learning brings challenges to solve Equation (2) directly.

*4.2. Our Fairness-Aware Approach*

We present our approach to solve the decentralized learning optimization problem, including the consideration of learning fairness, the design rationale, and our workflow. In the decentralized learning, the local learner trains the learning model with its dataset. The learning objective is determined by its local preference. To obtain the local learning model from other nodes, the local learner needs to find a learning pair by a peer-to-peer manner. In this regard, the optimization goal of each local learner can be considered as:

$$\mathbf{w}_i^{t+1} = arg\ minF_i(Agg(\mathbf{w}_i^t, \mathbf{w}_j^t), D_i), \forall\ j \in N. \tag{3}$$

Here, $Agg(\cdot)$ and $t$ are the aggregation scheme and the learning pair (iteration), respectively. Moreover, the model aggregation, in most previous works, adopts the weight averaging-based algorithms [16]. Such aggregation schemes could degrade the learning performance, if the local datasets are significantly statistical heterogeneous. For example, a number of local nodes with non-i.i.d. data distribution and the extreme massive local dataset might dominate the training process, resulting that the learning model benefits partial participants.

To address such a learning fairness issue, we design an adaptive local aggregation scheme which satisfies the following requirements: (i) the proposed aggregation scheme should be adaptive and dynamic based on the learning performance; (ii) the training participants have different local learning objectives with their own preferences; (iii) the proposed aggregation scheme should be lightweight, not bringing heavy computing load to the local learner.

In detail, our aggregation scheme is as follows:

$$P_i^t = \frac{pef_{\mathbf{w}_i, D_i}^t - pef_{\mathbf{w}_j, D_i}^t}{2 * pef_{max}} + C, \tag{4}$$

where $P_i^t$ is the weight of learning model $\mathbf{w}_i$ in the learning pair $t$, $pef_{\mathbf{w}_i, D_i}^t$ is the learning performance of learning model $\mathbf{w}_i$ over the dataset $D_i$, and $pef_{max}$ is the best learning performance. Note that we have multiple performance indicators for the learning performance $pef$ in Equation (4) (accuracy, loss value, etc.). As demonstrated in Equation (4), the proposed aggregation scheme allows each local learner to determine the aggregation weights with local preferences ($D_i$). To determine the weight of learning model $\mathbf{w}_j$ on the learner $i$, we have

$$P_{i,j}^t = 1 - P_i^t. \tag{5}$$

We show the details of our approach in Algorithm 1. Particularly, each local client finds a learning pair with a peer-to-peer manner iteratively, where the local learning objective is

determined by the device owner. To generate a local update, all clients adopt a gradient descent-based mechanism [33]. For Line 4 to Line 7, it shows that (i) a client requests the local learning model from its learning pair; (ii) the client evaluates the learning performance of received learning model; and (iii) it adaptively assigns the weight of the received model based on its learning performance (using Equations (4) and (5)). The complexity of our proposed algorithm on the client side is controlled by the number of learning pairs subject to available resources. Note that we assume each local client is capable of finding a learning pair in the training process by using technologies such as 5G, IoT search engine [34].

---

**Algorithm 1** The *Local Training* Procedure on the *Client*

---

  1: Launch the machine learning task
  2: Generate the local learning model $\mathbf{w_i}$
  3: Local training: $\mathbf{w}_i = arg\ min\ F_i(\mathbf{w}_i, D_i)$
  4: **if** *Finding a local learning pair* **then**
  5:      Request the learning model $\mathbf{w}_j$ from node $j$
  6:      Evaluate the performance of model $\mathbf{w}_j$ on dataset $D_i$
  7:      Generate the weight for $\mathbf{w}_i$ and $\mathbf{w}_j$ (Equations (4) and (5))
  8:      Local aggregation to find $\mathbf{w}_i^{t+1}$
  9: **else**
10:      Start next round local training
11: **end if**
12: Note that we adopt local accuracy as *pef* in the experiment.

---

## 5. Performance Evaluation

Extensive experiments have been conducted to validate the efficacy of our approach in comparison with the existing approaches. We first introduce the implementation details, including datasets, the deep learning models, baseline comparison approaches, and performance metrics/indicators, along with the parameters in our experiments. We then present the experimental results of our approach in comparison with several representative approaches and further discuss the performance results based on the performance metrics.

### 5.1. Methodology

**Deep learning models and datasets:** Two deep learning models are adopted to evaluate the proposed approach. We first consider a 5-layer Convolutional Neural Network (CNN) model (model 1) that consists of a couple $3 \times 3$ convolutional layers (32 channels for layer 1 and 64 channels for layer 2). A $2 \times 2$ maximum pooling layer is added after each convolutional layer, along with a fully-connected 512-unit layer and the *ReLu* activation function, and finally a softmax output layer (1,663,370 parameters) [35]. We then have a 7-layer CNN (model 2). To be specific, we adopt: (i) two $3 \times 3$ 64-channel convolutional layers that are followed by a $2 \times 2$ maximum pooling layer, (ii) two $3 \times 3$ 128-channel convolutional layers that are followed by a $2 \times 2$ maximum pooling layer, and (iii) three $3 \times 3$ 256-channel convolutional layers which are followed by a $2 \times 2$ maximum pooling layer. After that, the two fully connected layers with 512 units and *ReLu* activation are adopted [16].

Two public datasets are used in our study, E-MNIST [36] and CIFAR-10 [37]. Specifically, **E-MNIST** contains 280,000 characters over 10 balanced classes, including 240,000 training samples and 40,000 testing samples, in which each image has $28 \times 28$ pixels. **CIFAR-10** consists of 60,000 10-class images. We set 50,000 images for training and 10,000 images for testing. Both datasets provide rich data space so that our experimental setup can sample local datasets with diverse conditions (case 1 to case 4). In addition, E-MNIST dataset and CIFAR-10 have been widely used in existing related works such as [16]. All samples come from 10 balanced classes such as airplanes, cars, and others. Each sample is a $32 \times 32$ colorful image with the RGB channels with the size 3072 bytes, including 1024 bytes on the red, green, and blue channel values. Note that we train model 1 and model 2 over E-MNIST and CIFAR-10, respectively.

**Baseline approaches:** Two baseline approaches are considered for comparison: (i) *decentralized learning with full averaging aggregation (DL-Avg)* in which each learning pair aggregates the received model with the weight averaging algorithm [16] and (ii) *decentralized federated learning with segmented aggregation (DFL-Seg-Avg)* in which it adopts a model segment level decentralized federated learning that selects a set of model parameters to aggregate each segment [38].

**Performance metrics and scenarios:** The following performance metrics are used to evaluate the efficacy of our approach: (i) *Average local accuracy* in which each learning model over its local testing dataset is validated for the local accuracy and (ii) *accuracy distribution* in which the histograms of learning performance (i.e., local accuracy) are measured for the fairness of decentralized learning process. Note that the local learners determine the learning objective with its own preference.

Our experiments consider the various experimental scenarios, including balanced-i.i.d., unbalanced-i.i.d., balanced-non-i.i.d., and unbalanced-non-i.i.d. environments for local learners: (i) *i.i.d. data*—each local learner directly collects the local data samples from E-MNIST (CIFAR-10) dataset; (ii) *non-i.i.d. data*—each local learner collects $\zeta$ ($\zeta = 2, 3, 4$) classes from E-MNIST (CIFAR-10) dataset; (iii) *balanced data*—the size of local dataset is 800 (1000) images from E-MNIST (CIFAR-10) dataset; (iv) *unbalanced data:* the size of local dataset ranges from 400 to 1600 images. Moreover, we assign 2 nodes with the large dataset (10,000 samples) in the unbalanced-non.i.i.d. scenario. Table 2 lists the key parameters in our evaluation.

**Table 2.** List of parameters.

| Parameter | Value |
| --- | --- |
| Total node number | 20 |
| Local epoch | 2 |
| Non-i.i.d. coefficient $\zeta$ | 2, 3, 4 |
| Minibatch size | 100 |
| Learning pair | 1000/2000 |
| Total scenarios | 4 |

*5.2. Experimental Results*

We first validate our approach and two baseline approaches in the balanced-i.i.d. data scenario. Figure 3 shows the accuracy distribution of all three approaches. As seen in the figure, all of the approaches achieve the acceptable performance where the learning performance of the majority of the participants is higher than 90%. This is because a balanced-i.i.d. data distribution scenario eliminates the diversity of local learners, which makes all of the local learners become homogeneous. Similar to Figure 3, we obtain the accuracy distribution of all of the approaches after we have 2000 local learning pairs on the CIFAR-10 dataset. As shown in Figure 4, the performance gap of our approach and two baseline approaches is not significant. On the other hand, over half of the local learners could achieve the acceptable performance which is between 60% to 75%. In Figures 5 and 6, we demonstrate the evaluation results in the unbalanced-i.i.d. data distribution scenario. Both of them show that all of the approaches could achieve a comparable learning performance in a homogeneous learning scenario.

Compared to case 1 (balanced-i.i.d. data distribution) and case 2 (unbalanced-i.i.d. data distribution), we consider the decentralized learning processes in case 3 (balanced-non.i.i.d.) and case 4 (unbalanced-non.i.i.d.) for statistical heterogeneity. As shown in Figure 7, noticeably over half of the local learners reach 90% (local accuracy) on the E-MNIST dataset. Figure 8 shows that our approach achieves the best learning performance among all of the tested approaches where 15 nodes reach 55% (local accuracy) on the CIFAR-10 dataset.

Additionally, we record the evaluation results of case 4 in Figures 9 and 10. Figure 9 shows that our approach performs the best among all of the tested approaches where 14 nodes reach 80% (local accuracy) on the E-MNIST dataset. When it comes to the CIFAR-10 dataset, our approach, as seen in Figure 10, shows a great capability, which not only successfully makes eight local learners to achieve the acceptable learning performance but also avoids resulting in local learners with an extremely low performance (30% to 45%). There are a number of reasons that lead to these evaluation results. We find that the decentralized learning with a statistical heterogeneous setting brings the inevitable challenge which degrades the learning performance for all of the tested approaches. To strengthen the local learning objectives, our approach aggregates the local learning model with the consideration of its learning performance. In case 4 (unbalanced-non.i.i.d. scenario), we assign two nodes with the massive local dataset (10,000 samples), resulting in these two nodes bringing negative impacts on other learning pairs. By using Equations (4) and (5), our proposed fairness-aware approach allows each node to train the local learning model with its own learning preference and objective. To be specific, in local aggregations, each local node evaluates the received learning model based on its local learning objective to determine the weight of aggregation. In this sense, the experimental results, in case 4 (unbalanced-non.i.i.d. scenario), clearly validate that our approach eliminates the negative impact of statistical heterogeneity (both the non-i.i.d. data distribution and unbalanced dataset) and that it outperforms the baseline approaches. Furthermore, we present numerical results in Tables 3–5, showing that our approach achieves the best performance under most cases (case1-E-MNIST, case3-E-MNIST, case4-E-MNIST, case2-CIFAR-10, case3-CIFAR-10, and case4-CIFAR-10).

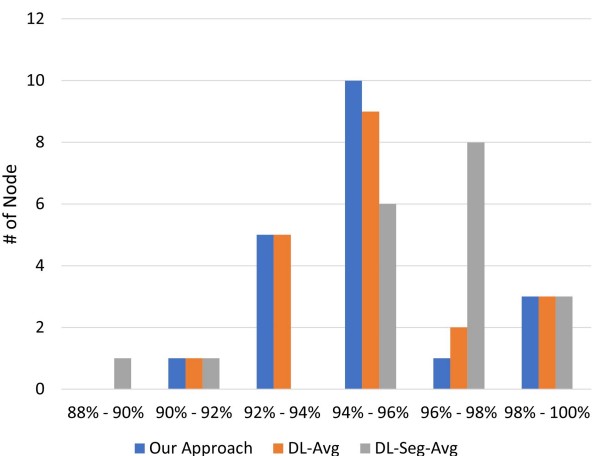

**Figure 3.** Accuracy Distribution for Case 1 (E-MNIST).

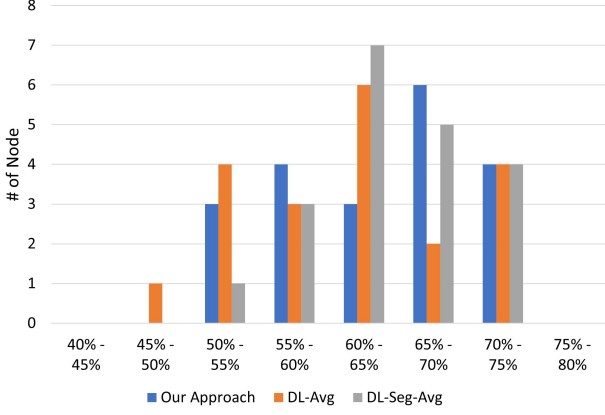

**Figure 4.** Accuracy Distribution for Case 1 (C-10).

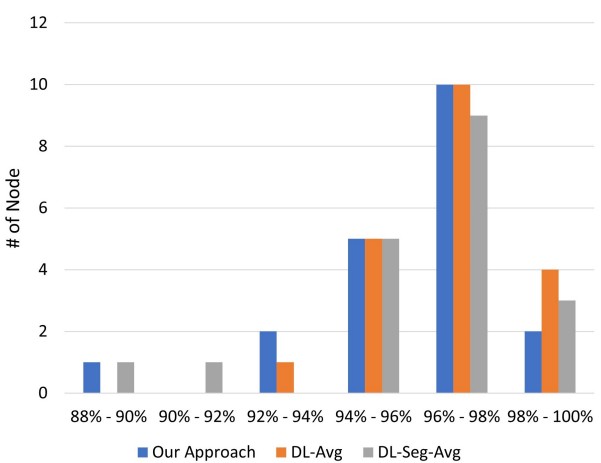

**Figure 5.** Accuracy Distribution for Case 2 (E-MNIST).

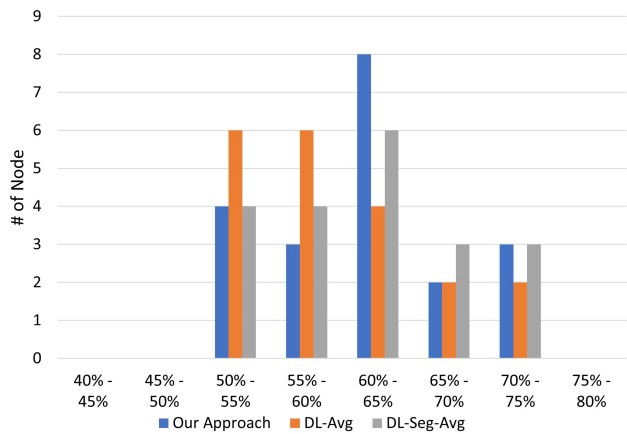

**Figure 6.** Accuracy Distribution for Case 2 (C-10).

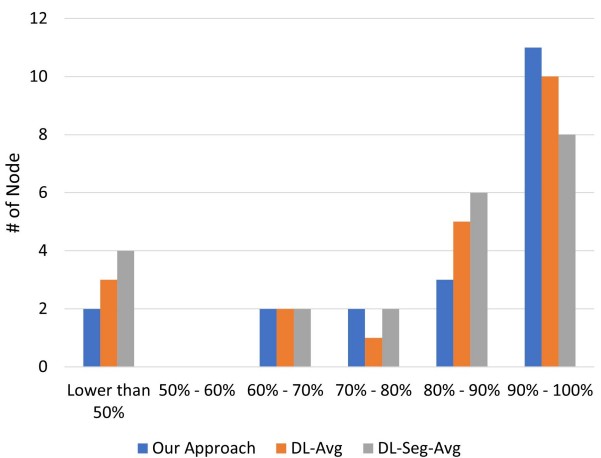

**Figure 7.** Accuracy Distribution for Case 3 (E-MNIST).

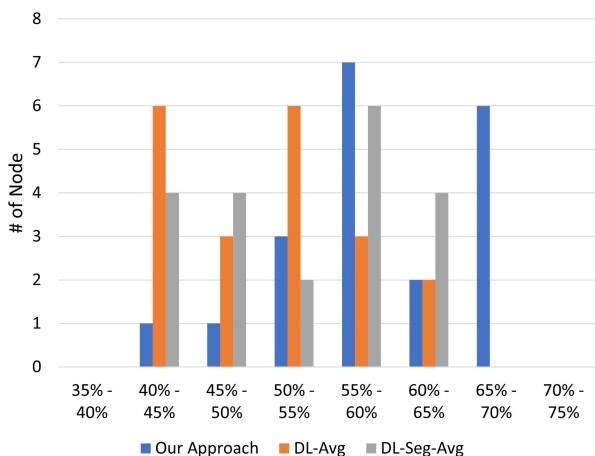

**Figure 8.** Accuracy Distribution for Case 3 (C-10).

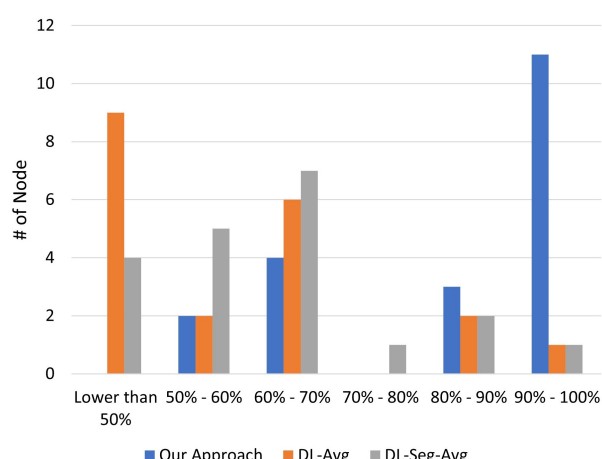

**Figure 9.** Accuracy Distribution for Case 4 (E-MNIST).

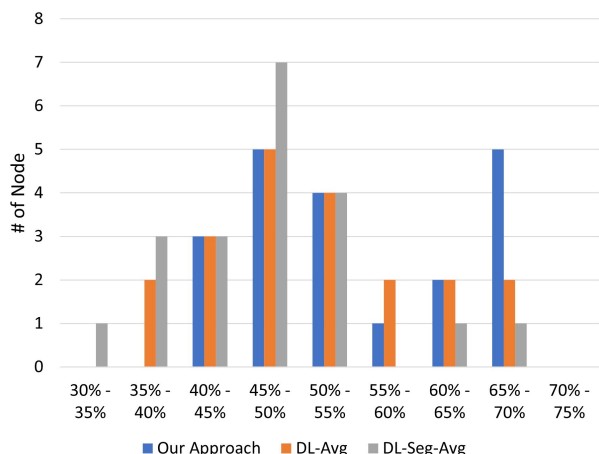

**Figure 10.** Accuracy Distribution for Case 4 (C-10).

**Table 3.** Final results (local average accuracy).

|  | Our Approach | DL-Avg | DFL-Seg-Avg |
|---|---|---|---|
| **Case#1 (E-MNIST)** | **95.61%** | 95.07% | 94.84% |
| **Case#2 (E-MNIST)** | 95.60% | 96.29% | 95.84% |
| **Case#3 (E-MNIST)** | **84.10%** | 81.68% | 82.08% |
| **Case#4 (E-MNIST)** | **85.10%** | 53.58% | 60.83% |
| **Case#1 (CIFAR-10)** | 63.03% | 61.78% | 64.36% |
| **Case#2 (CIFAR-10)** | **62.37%** | 60.51% | 61.32% |
| **Case#3 (CIFAR-10)** | **58.69%** | 53.77% | 52.47% |
| **Case#4 (CIFAR-10)** | **54.79%** | 48.74% | 45.80% |

**Table 4.** Number of nodes in various accuracy distributions (E-MNIST).

| | Numerical results in Case 1 (E-MNIST) | | |
|---|---|---|---|
|  | **LA below 92%** | **LA over 92%** | **LA over 96%** |
| **Our Approach** | 1 | 19 | 4 |
| **DL-Avg** | 1 | 19 | 5 |
| **DFL-Seg-Avg** | 2 | 18 | 11 |
| | Numerical results in Case 2 (E-MNIST) | | |
|  | **LA below 92%** | **LA over 92%** | **LA over 96%** |
| **Our Approach** | 1 | 19 | 12 |
| **DL-Avg** | 0 | 20 | 14 |
| **DFL-Seg-Avg** | 2 | 18 | 12 |
| | Numerical results in Case 3 (E-MNIST) | | |
|  | **LA below 80%** | **LA over 80%** | **LA over 90%** |
| **Our Approach** | **6** | **14** | **11** |
| **DL-Avg** | 6 | 14 | 10 |
| **DFL-Seg-Avg** | 8 | 12 | 8 |
| | Numerical results in Case 4 (E-MNIST) | | |
|  | **LA below 80%** | **LA over 80%** | **LA over 90%** |
| **Our Approach** | **6** | **14** | **11** |
| **DL-Avg** | 17 | 3 | 3 |
| **DFL-Seg-Avg** | 16 | 4 | 3 |
| | LA = (Local Accuracy) | | |

**Table 5.** Number of nodes in various accuracy distributions (CIFAR-10).

| | LA below 55% | LA over 55% | LA over 65% |
|---|---|---|---|
| Numerical results in Case 1 (CIFAR-10) | | | |
| Our Approach | 3 | 17 | 10 |
| DL-Avg | 5 | 15 | 6 |
| DFL-Seg-Avg | 1 | 19 | 9 |
| Numerical results in Case 2 (CIFAR-10) | | | |
| Our Approach | 4 | 16 | 5 |
| DL-Avg | 6 | 14 | 4 |
| DFL-Seg-Avg | 4 | 16 | 6 |
| Numerical results in Case 3 (CIFAR-10) | | | |
| Our Approach | 5 | 15 | 6 |
| DL-Avg | 15 | 5 | 0 |
| DFL-Seg-Avg | 10 | 10 | 0 |
| Numerical results in Case 4 (CIFAR-10) | | | |
| Our Approach | 12 | 8 | 5 |
| DL-Avg | 14 | 6 | 2 |
| DFL-Seg-Avg | 18 | 2 | 1 |
| LA = (Local Accuracy) | | | |

## 6. Discussion

The experimental results in Section 5 confirm that the fairness-aware approach achieves an exceeding performance over multiple scenarios. However, there are some open issues in this field for further investigation. For example, in this paper, we assume that all of the local learners are honest and that there are no malicious nodes. The impact of malicious nodes which disseminate stealthy backdoor attacks in the training process could draw a new research direction to improve the learning integrity. In what follows, we briefly discuss several open issues.

### 6.1. Federated Learning Heterogeneity Issues

With the advance and deployment of IoT systems, distributed machine learning has attracted great attention [39]. Federated learning, one of the representative distributed machine learning technologies, tends to train a global learning model with a group of local clients. Federated learning adopts a central parameter to organize the training process, including the model aggregation and model update. To eliminate the privacy concern, none of the nodes are allowed to access other local datasets. On the other hand, federated applications, such as the G-board, show their great potential to embrace distributed IoT devices [40].

However, the inherent heterogeneity of IoT devices becomes the emerging bottleneck to deploy the federated learning framework under a large-scale real-world scenario. The inherent heterogeneity of IoT devices could be cataloged as three aspects, which include non-i.i.d. data distribution, unbalanced data, and heterogeneous devices. In this stage, there are some research efforts focusing on training a global learning model with non-i.i.d. local clients. For example, Zhang et al. [21] investigated the learning performance

of federated learning under non-i.i.d. data distribution. Authors first showed that the accuracy of federated learning degrades significantly with highly skewed non-i.i.d. data. Moreover, authors analyzed this accuracy reduction that could be evaluated via the earth mover's distance (EMD).

We consider unbalanced data and heterogeneous devices to still lead unsolved challenges. Unbalanced data cause the learning fairness in federated learning. For example, the global learning would be dominated by the local clients with a massive dataset, ignoring the learning objectives of the relatively small nodes. Heterogeneous IoT devices bring the extravagant communication cost into federated learning. This is because the lack of a compatible communication protocol confines the deployment of federated learning over a real-world scenario. Thus, the design of compatible communication protocols requires further research.

### 6.2. Decentralized Learning and Transfer Learning

Decentralized learning and transfer learning show great potential to play a critical role in IoT systems. Decentralized learning allows local clients to conduct the training without a central parameter server [41]. To exchange the model updates, local clients would send their local updates via a peer-to-peer manner. On the other hand, transfer learning presents a viable tool to leverage the existing (well-trained) machine learning models in the networking system [42]. In this sense, we consider that the development of an integrated framework is necessary so that both decentralized learning and transfer learning technologies can be integrated. For instance, Ma et al. [43] proposed a decentralized learning scheme that integrates both decentralized learning and transfer learning technologies, namely decentralized learning via adaptive distillation (DLAD). The proposed scheme adopts knowledge distillation to transfer the knowledge of mature teacher models into student models. Moreover, to train a learning model with an unlabeled dataset, the proposed scheme adaptively emphasizes those with higher confidence on given inputs.

### 6.3. Trustworthiness in Decentralized Learning

In decentralized learning, all of the local clients aim to train their local model with a peer-to-peer manner. To increase the learning performance, some research efforts combine both decentralized learning and transfer learning technologies. For example, Li et al. [16] addressed the problem of decentralized learning in IoT systems. To overcome slow convergence and degraded learning performance, the proposed approach introduces a mutual knowledge transfer algorithm into a decentralized learning framework, where a set of IoT devices optimizes the learning model without a central server. Nonetheless, the lack of consideration for a trustworthy evaluation may result in potential security threats and further reduce accuracy, slow convergence, and the dissemination of unexpected backdoors. As a possible solution, we shall systematically study the techniques to ensure the security trust in the decentralized learning process so that the security risk of all of the components (e.g., model updates, local clients) can be modeled and assessed. By doing this, it enables each task aggregator to collect trust scores from each learning client.

On the other hand, the evaluation of existing models remains an unsolved issue. We consider the future research works cataloged as two aspects. First, we shall methodically investigate the dissemination of malicious backdoors, which are delivered by some mature (well-trained) learning models. Second, we shall develop the defensive mechanisms to detect the potential backdoors. For instance, the trusted evaluation may record behaviors in both short-term and long-term aspects from each decentralized learning with mechanisms such as a blockchain-based consensus mechanism. Such information can significantly help each client prioritize the received information from other nodes so that the robustness of the decentralized learning framework can be improved.

Similar to the trustworthy evaluation, we shall conduct further research efforts on designing an incentive scheme in federated (decentralized) learning systems. A proper incentive scheme reveals the relationship between the cost of the training process and

the profit of sharing updates. The incentive model can help theoretical models to assess the different effects of node behaviors (both benign and malicious nodes) and further improve the robustness of the decentralized learning framework. Last but not least, we notice that the match mechanism of decentralized learning also remains an open issue. To address this problem, we should fully study the integration of the state-of-the-art networking communication and computing technologies and decentralized learning on an application level.

## 7. Final Remarks

In this paper, the issue of decentralized machine learning in IoT systems has been addressed. In particular, we have first designed a framework to explore the problem space of deploying decentralized machine learning in IoT systems. Based on the designed framework, we have then studied a specific problem, i.e., distributed machine learning under a statistical heterogeneous scenario, in which the learning nodes train the model with unbalanced and non-i.i.d. local datasets. To tackle the issue, we have designed a peer-to-peer decentralized learning approach that addresses the learning fairness problem and the inherent statistical heterogeneity in the training process. The proposed adaptive aggregation scheme iteratively updates local weights based on the run-time learning performance, advocating for learning fairness by avoiding small-group domination among local learners. The empirical evaluation results have confirmed the efficacy of our approach and demonstrated that our approach benefits the majority of local learners in heterogeneous settings, in comparison with several baseline approaches. Furthermore, we have discussed some open issues for future research based on our study.

**Author Contributions:** Conceptualization, Z.C. and W.L.; methodology, Z.C., W.L. and W.Y.; software, P.T. and Q.W.; validation, P.T. and W.Y.; formal analysis, Z.C. and W.L.; data curation, Z.C.; writing—original draft preparation, Z.C., Q.W. and W.Y.; writing—review and editing, Q.W. and W.Y.; problem space and formalization, Q.W. and W.Y.; visualization, Z.C.; supervision, W.L. and W.Y.; project administration, W.L. and W.Y. All authors have read and agreed to the published version of the manuscript.

**Funding:** This research was supported by the Jess and Mildred Fisher Endowed Professorship of Computer Science from the Fisher College of Science and Mathematics, Towson University. Any opinions, findings, and conclusions or recommendations expressed in this material are those of the author(s) and do not necessarily reflect the views of the funding source.

**Institutional Review Board Statement:** Not applicable.

**Informed Consent Statement:** Not applicable.

**Data Availability Statement:** Not applicable.

**Conflicts of Interest:** The authors declare no conflict of interest.

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
