# Peer review of "A Fairness-Aware Peer-to-Peer Decentralized Learning Framework with Heterogeneous Devices"

_futureinternet, doi:10.3390/fi14050138_

Round 1

Reviewer 1 Report

The paper is well organized. 

A few minor issues are as follows:

  • Algorithm 4.2 is not given. Please check for missing items in the paper.
  • Figure 3-10 should appear within section 5.
  • please provide detailed analysis on the results. Current status of experiment result touches only the visible results, but it would be much appreciated if the authors could provide reasons it is performing better than the other compared methods.

Author Response

Manuscript ID: futureinternet-1704106

Journal: Future Internet

Title: A Fairness-aware Peer-to-peer Decentralized Learning Framework with Heterogeneous Devices

Decision: Minor revision

Authors: Zheyi Chen, Weixian Liao, Pu Tian, Qianlong Wang, and Wei Yu

Dear Ms. Tracy Peng and Reviewers,

We would like to again thank you all for your valuable and insightful comments, which have contributed to further improving the quality of the paper. We have thoroughly revised our paper in response to Reviewer 1’s and Reviewer 2’s comments. We have also further improved the presentation of our paper. The key changes made to the previous submission are highlighted in blue color in the revised paper.

Thank you very much!

Best Regards,

Zheyi Chen, Weixian Liao, Pu Tian, Qianlong Wang, and Wei Yu

Reviewer 2 Report

  1. What are the reasons behind choosing E-MNIST and CIFAR-10 datasets? Please add some explanation.
  2. What are the limitations of this framework?

Author Response

(The authors gave the same response as above.)
